# Growth Retardation in the Course of Fanconi Syndrome Caused by the 4977-bp Mitochondrial DNA Deletion: A Case Report

**DOI:** 10.3390/children8100887

**Published:** 2021-10-04

**Authors:** Ting Li, Zhihong Lu, Jingjing Wang, Junyi Chen, Haidong Fu, Jianhua Mao

**Affiliations:** Department of Nephrology, The Children’s Hospital, Zhejiang University School of Medicine, National Clinical Research Center for Child Health, National Children’s Regional Medical Center, 3333 Binsheng Road, Hangzhou 310052, China; liting170209@126.com (T.L.); lzhsjf@zju.edu.cn (Z.L.); 6506027@zju.edu.cn (J.W.); chenjunyi0578@163.com (J.C.); ch61312@163.com (H.F.)

**Keywords:** mitochondrial DNA, 4977-bp deletion, Fanconi syndrome, growth retardation, children

## Abstract

Fanconi syndrome is one of the primary renal manifestations of mitochondrial cytopathies caused by mitochondrial DNA (mtDNA) mutation. The common 4977-bp mtDNA deletion has been reported to be associated with aging and diseases involving multiple extrarenal organs. Cases of Fanconi syndrome caused by the 4977-bp deletion were rarely reported previously. Here, we report a 6-year-old girl with growth retardation in the course of Fanconi syndrome. She had mild ptosis and pigmented retinopathy. Abnormal biochemical findings included low-molecular-weight proteinuria, normoglycemic glycosuria, increased urine phosphorus excretion, metabolic acidosis, and hypophosphatemia. Growth records showed that her body weight and height were normal in the first year and failed to thrive after the age of three. Using a highly sensitive mtDNA analysis methodology, she was identified to possess the common 4977-bp mtDNA deletion. The mutation rate was 84.7% in the urine exfoliated cells, 78.67% in the oral mucosal cells, and 23.99% in the blood sample. After three months of oral coenzyme Q10 and levocarnitine treatment in combination with standard electrolyte supplement, her condition was improved. This is a report of growth retardation as the initial major clinical presentation of Fanconi syndrome caused by the deletion of the 4977-bp fragment. Renal tubular abnormality without any other extrarenal dysfunction may be an initial clinical sign of mitochondrial disorders. Moreover, considering the heterogeneity of the phenotypes associated with mtDNA mutations, the risk of developing Kearns–Sayre syndrome (KSS) with age in this patient should be noted because she had ptosis, retinal involvement, and changes in the brain and skeletal muscle.

## 1. Introduction

Fanconi syndrome is a rare disorder that occurs due to the overall dysfunction of proximal tubules of the kidney, resulting in impaired reabsorption and excessive urinary excretion of glucose, amino acids, uric acid, bicarbonates, phosphates, and other solutes. In severe cases, these excessive losses lead to electrolyte imbalance, dehydration, acidosis, osteomalacia, rickets, and growth failure. Numerous inherited or acquired conditions cause Fanconi syndrome, and mitochondrial cytopathy is one of the inherited conditions associated with Fanconi syndrome [1].

Mitochondrial cytopathies are a heterogeneous group of abnormalities characterized by damaged oxidative phosphorylation. The organs that are highly dependent on mitochondrial energy are particularly prone to be affected. As the kidney has a great energy requirement to facilitate glomerular filtrate reabsorption, it is one of the organs that can be easily affected by mitochondrial abnormalities. The most common renal manifestation that is associated with mitochondrial cytopathy is Fanconi syndrome. Bartter syndrome, focal segmental glomerulosclerosis (FSGS), and tubulointerstitial nephropathy are also seen with mitochondrial cytopathy. The mitochondrial cytopathies originate from mutations of the genes in nuclear DNA, which encodes mitochondrial proteins, or in mitochondrial DNA (mtDNA) [2]. Approximately 10–15% of pediatric mitochondrial disorders occur as the result of mutations of the genes in the mtDNA.

Here, we report a girl who had the common 4977-bp deletion mutation from the nucleotide position 8470 to 13,446, presenting with proximal renal tubulopathy as the first sign, accompanied by growth retardation, ptosis, pigmented retinopathy, and abnormalities in the brain and skeletal muscle.

## 2. Case Presentation

The girl was admitted to our hospital at the age of 6 years because she had vomiting and diarrhea for one week. She had been diagnosed with severe malnutrition and Fanconi syndrome one year before admission and was prescribed potassium citrate, disodium hydrogen phosphate/sodium dihydrogen phosphate, and magnesium supplementation. However, the blood magnesium and phosphorus levels were close to but still below the normal range, there was no weight gain during the one-year treatment period, and the height increased by 3 cm. Currently, the girl’s length is 105 cm (less than 2SD) and weight is 12 kg (less than 3SD). The growth curve is shown in Figure 1. Her body weight and height were normal in the first year of life, and the growth retardation aggravated after the age of three. She also had exercise intolerance and a history of recurrent upper respiratory tract infection and diarrhea. All medication was discontinued for one week due to severe gastrointestinal distress. Clinical examination demonstrated normal intelligence but severe malnutrition, right eye ptosis, and exercise intolerance. Routine blood chemistry revealed metabolic acidosis (pH 7.223; PCO2 17 mmHg; Bicarbonate 6.8 mmol/L; Anion gap 21.3 mmol/L; Lactate 8.9 mmol/L) and the blood levels of phosphorus (1.01 mmol/L (1.29–2.26 mmol/L)), magnesium (0.4 mmol/L (0.73–1.06 mmol/L)), and uric acid (94 μmol/L (155–357 μmol/L)) were low. There was normal renal function (serum creatinine 46 μmol/L). Urinalysis revealed a generalized dysfunction of the proximal tubule with low-molecular-weight proteinuria, normoglycemic glycosuria (urine sugar +++), and increased uric acid (uric acid excretion fraction 44.6%), magnesium (113.4 mg/1.73 m^2^/24 h), and phosphorus/creatinine (2.22 mg/mg) in urine excretion. The protein was 204.4 mg in 24 h urine sample. Magnetic resonance imaging (MRI) of the brain showed symmetrical abnormal signals in the brain stem, and pigmentation was seen upon fundus examination (Figure 2). The electromyography demonstrated myogenic damage. The dual-energy X-ray showed low bone mass (Z-score: −3.9). There were no obvious abnormalities on cardiac color Doppler ultrasound and electrocardiogram. She had no sensorineural hearing loss, ataxia, tremor, or cognitive dysfunction. The mother denied that the child had a long history of medication use before the age of 5. Other causes of Fanconi syndrome, such as genetic metabolic diseases—cystinosis, Lowe syndrome, hepatolenticular degeneration, and glycogen disease—were ruled out by physical examination, laboratory testing, and next-generation sequencing (NGS), and no other significant mutations were found by NGS. However, the mtDNA sequencing showed the 4977-bp fragment deletion (nt8470-nt13446), but the mutation rate of mtDNA in the blood sample was only 23.99%. Then, mtDNA from the oral mucosal cells and exfoliated cells in urine was also used. The mutation rate was 84.7% in the urine exfoliated cells and 78.67% in the oral mucosal cells, implicating that this mitochondrial deletion may have occurred de novo in the oocyte or at a very early stage of embryogenesis. 

The mother denied any movement disorder, intellectual abnormality, or growth retardation in other family members. No abnormalities were found in the results of routine urinalysis, blood chemistry testing, and mtDNA sequence from the grandmother, mother, and brother of the patient.

After establishing the diagnosis, the patient was administrated with coenzyme Q10 100 mg/d and levocarnitine 1 g/d to improve the mitochondrial function in combination with standard electrolyte supplementation. Blood phosphorus and magnesium levels slowly recovered to normal levels in one month (Phosphorus: 1.34 mmol/L; Magnesium: 0.79 mmol/L). After three months of treatment, the exercise intolerance was gradually alleviated.

## 3. Mitochondrial DNA Analysis

The samples used were from the blood, oral mucous membrane, and morning urine. The extraction of mtDNA was performed using a mtDNA extraction kit. The full-length mtDNA was amplified using PCR with high-fidelity DNA polymerase. The amplified mtDNA was separated by agarose gel electrophoresis and purified using a DNA gel extraction kit. Genomic DNA was sheared to approximately 200 bp fragments using the Covaris sonicator. A DNA end-repairing agent was used for blunting and phosphorylation of DNA ends. Adding an adenine to the 3′ end of the repaired blunt-end products was performed by the following ligation reaction. The ligation of the adapter at the A-tailing end was catalyzed by a T4 DNA ligase (Thermo Fisher Scientific, Eugene, OR, USA). The ligated DNA products were amplified through 4-6 rounds of LM-PCR. Magnetic beads were used to purify the PCR products. The length of the inserted fragments was detected using the Agilent 2100 Bioanalyzer, and the effective concentration was quantified by qPCR. The PE150 (paired-ended 150 bp) sequencing was done using the NovaSeq 6000 sequencing system. Clean data were obtained by quality control and removing low-quality data. The sequenced data were aligned to the reference sequence NC_012920 (human complete mitochondrial genome 16,569 bp circular DNA) using the Burrows-Wheeler Aligner (BWA) software. SNPs and indels were called using SAMtools and Pindel software packages, respectively. The depth and quality of reads were adjusted to screen the reliable variants. The variants were mapped to the reference mutations to find matches in the MITOMAP human mitochondrial genome database. The “confirmed pathogenic” and “likely pathogenic” variants were screened according to the MITOtip.

## 4. Discussion

The diagnosis in the patient was confirmed as Fanconi syndrome, which was related to the 4977-bp deletion. The specific mtDNA deletion of 4977 bp occurred between two 13-bp direct repeats in the mtDNA sequence, at nucleotide positions between 8470 and 13459 [3]. The deletion contains the genes encoding four polypeptides for complex I (ND3, ND4, ND4L, and ND5), one for complex IV (COX3), two for complex V (ATP8, ATP6), and five tRNA genes for the amino acids G, R, H, S, and L. The 4977-bp deletion is the most common deletion among more than 90 large-scale deletions of human mtDNA that are associated with aging and mitochondrial myopathies, which can lead to three related mtDNA diseases: Pearson syndrome, Kearns–Sayre syndrome (KSS), and chronic progressive external ophthalmoplegia (CPEO).

The mitochondrial mutations reported in patients with Fanconi syndrome are listed in Table 1, obtained by searching the PubMed database. There was only one case of 4977-bp deletion reported by Niaudet et al. [4] and the girl was diagnosed with Pearson’s syndrome before the age of 2 years and had Fanconi syndrome at the age of 3 years and 9 months. She had no external ophthalmoplegia, pigmentary retinopathy, or muscle weakness. However, our patient had Fanconi syndrome as the first symptoms at the age of 5 years without the clinical manifestation of Pearson syndrome; this is a rare report of growth retardation as the initial major clinical manifestation of Fanconi syndrome caused by the deletion of the 4977-bp fragment. Moreover, since proximal tubule cells are highly dependent on ATP molecules, renal manifestations without any other extrarenal dysfunction may be the first clinical symptom of mitochondrial disorders.

The mtDNA deletion syndrome links to any case of a single mtDNA deletion, and the case may develop from one clinical syndrome to another over time. The three typical abnormal phenotypes caused by mtDNA deletions are Pearson syndrome, KSS, and progressive external ophthalmoplegia. For all mtDNA pathogenic mutations, the clinical manifestation is determined by three factors: heteroplasmy (relative abundance of the mutated mtDNA), threshold effect (tissue vulnerability to the impaired oxidative metabolism), and the tissue distribution of the mtDNA deletion [19]. As for this patient, the mitochondrial mutation rate in the renal cells was significantly higher than that in the other tissues, which may have been the main reason for the renal abnormality as the main manifestation. 

KSS is a rare mitochondrial DNA deletion syndrome diagnosed by the presence of onset at less than 20 years of age, ophthalmoplegia, pigmentary retinopathy, and one of the following: cerebellar syndrome, cerebrospinal fluid (CSF) protein above 100 mg/dL, or cardiac conduction defects. The KSS affects many organ systems, leading to encephalomyopathy, endocrinopathies, renal tubular diseases, and sensorineural hearing loss. Tzoufi et al. [11] reported a 5-year-old child with KSS due to a 9-kbp deletion who had the simultaneous presentation of short stature, Fanconi syndrome, palpebral ptosis, retinopathy, myopathy, Addison’s disease, primary hypoparathyroidism, and high CSF protein. Mihai et al. [13] reported an 18-year-old man who developed short stature, Fanconi syndrome, and palpebral ptosis from 4 years old, and external ophthalmoplegia, myopathy, cerebellar ataxia, retinitis pigmentosa, sensorineural hearing impairment, hyperaldosteronism, hypoparathyroidism, diabetes mellitus, and cardiac conduction defect from 9 years of age. The diagnosis of KSS was delayed for many years. Mori et al. [12] reported a girl with a 5.4-kbp deletion who developed short stature and anhidrosis at the age of 2, Fanconi syndrome at the age of 6, and hearing loss and impaired visual acuity at the age of 8. Since then, ptosis and external ophthalmoplegia gradually occurred. Moreover, she had disturbances of consciousness accompanied by vomiting at the age of 12, progressive myopathy, abnormal CSF, heart block, and retinopathy at the age of 13, and was diagnosed with KSS finally. In the latter two cases, Fanconi syndrome appeared a few years before the onset of KSS, indicating that patients with Fanconi syndrome as the first symptom need careful long-term follow-up. This patient had Fanconi syndrome, growth retardation, ptosis, retinopathy, abnormal brain signals on MRI, and muscle damage shown by electromyography, but no heart block, cerebellar ataxia, hearing impairment, or endocrine abnormalities. Although KSS cannot be diagnosed at present, we should be alert to the risk that this case may develop into KSS for timely prevention and intervention.

Another question that needs to be considered is the origin of the mutant mitochondria. We found no related symptoms in any family member and no mutations in the mtDNA from the mother’s blood, urine, or oral cells. This suggests that the mitochondrial mutations in the child may have originated from spontaneous mutations in the oocyte or at a very early stage of embryogenesis.

Further, it is worth noting that the 4977-bp deletion of the mtDNA is also the most prevalent and abundant one that has been associated with aging in humans [20]. The deletion, as a universal DNA marker of aging, has been extensively researched. Currently, it is believed that with the increase in age, the loss of mitochondrial 4977-bp occurs in normal adults with a higher incidence [21]. In this case, will the symptoms of the child become worse as she ages? Will the mutation be passed on to the next generation? All these unknown factors need to be followed up.

In conclusion, mitochondrial diseases with growth retardation as the first clinical symptoms of Fanconi syndrome caused by the mtDNA 4977-bp fragment deletion are rare. Renal tubular abnormality without any other extrarenal dysfunction may be the initial manifestation of mitochondrial disorders, which should be followed up in the long term. Moreover, we must pay attention to the possibility that this case may develop into KSS, so as to prevent and intervene in time. 

## Figures and Tables

**Figure 1 children-08-00887-f001:**
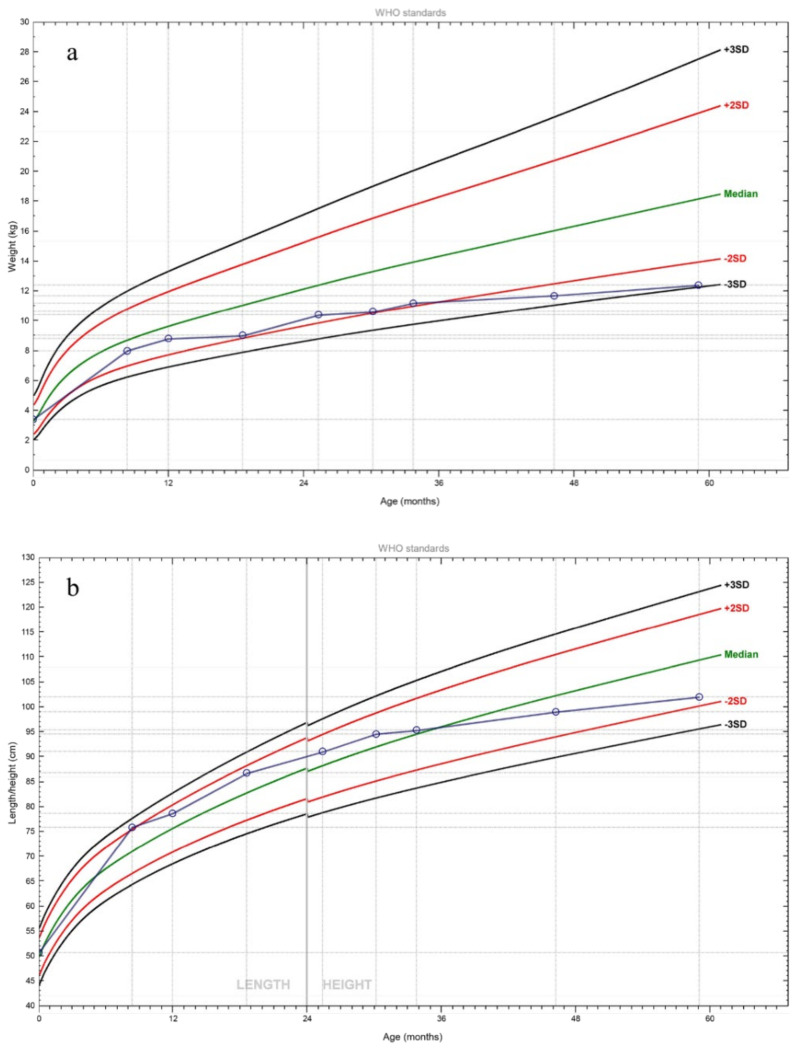
Growth charts for the child, which are shown as violet line: (**a**) growth curve for body weight; (**b**) growth curve for body length or height.

**Figure 2 children-08-00887-f002:**
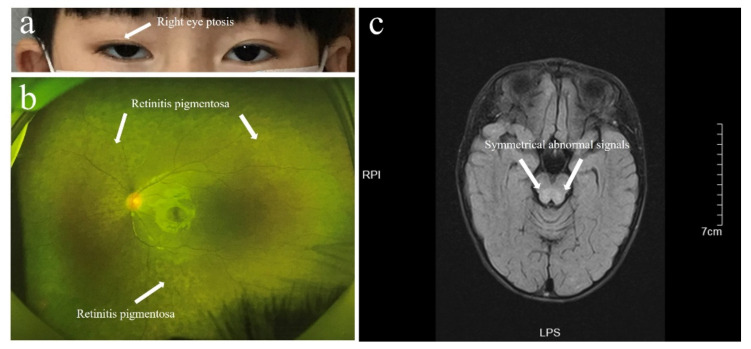
Abnormalities of the patient: (**a**) right eye ptosis; (**b**) retinitis pigmentosa; (**c**) head MRI examination shows symmetrical abnormal signals in the brain stem.

**Table 1 children-08-00887-t001:** Overview of mitochondrial mutations reported in patients with Fanconi syndrome.

Category	Mutation	Reference
Corneal clouding	7.4 kbp deletion	[5]
Isolated proximal tubular abnormalities	7.3 kbp deletion	[6]
Anemia	3.3 kbp deletion	[7]
Anemia and ptosis	2.8 kbp deletion	[8]
Retinitis pigmentosa	6.7 kbp deletion	[9]
Diabetes and muscle wasting	5 kbp deletion	[10]
Kearns–Sayre syndrome	9 kbp deletion	[11]
Kearns–Sayre syndrome	5.4 kbp deletion	[12]
Kearns–Sayre syndrome	A deletion in the mtDNA	[13]
Pearson syndrome	3.5 kbp deletion	[14]
Pearson syndrome	4.9 kbp deletion	[14]
Pearson syndrome	6.3 kbp deletion	[15]
Pearson syndrome	4977bp deletion	[4]
Pearson syndrome	5.7 kbp deletion	[16]
Mitochondrial encephalomyopathy with lactic acidosis and stroke-like episodes	6.0 kbp deletion	[17]
Multisystem diseases	3670 bp deletion	[18]

## Data Availability

The datasets used and/or analyzed during the current study are available from the corresponding author upon reasonable request.

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
