# Peer review of "Growth Retardation in the Course of Fanconi Syndrome Caused by the 4977-bp Mitochondrial DNA Deletion: A Case Report"

_children, 2021, doi:10.3390/children8100887_

Round 1

Reviewer 1 Report

Author show a very interesting case of a girl with Fanconi syndrome and mitochondrial DNA mutation. Although similar cases are very rare (Zhu YH, Zhang QJ, Wang QJ. Lin Chung Er Bi Yan Hou Tou Jing Wai Ke Za Zhi. Mitochondrial DNA deletion syndrome: a case report and literature review. 2019 Sep;33(9):808-813. doi: 10.13201/j.issn.1001-1781.2019.09.004; Lee JJ, Tripi LM, Erbe RW, Garimella-Krovi S, Springate JE. A mitochondrial DNA deletion presenting with corneal clouding and severe Fanconi syndrome. Pediatr Nephrol. 2012 May;27(5):869-72. doi: 10.1007/s00467-011-2096-2), presented manuscript indicates, that Fanconi syndrome diagnosis should raise our concerns about mitochondrial dysfunction.

I think this manuscript needs some improvement

1) please explain abbreviations when used for the first time - KSS (line 23 in the abstract);

2) proximal renal tubulopathy (proteinuria, glucosuria...) is rather a sign than a symptom of the disease (lines 47-48);

3) please add in Figure 1 caption information that patients results are shown as violet line;

4) please add arrows to Figure 2b and 2c, indicating pathological findings you describe;

5) I think that methodology about mtDNA isolation and testing should be placed under separate section (i.e. 'mitochondrial DNA analysis'), not in the case presentation itself;

6) What do you that ' 3 year 9 months old patient had abnormal renal tubules'? (lines 146)? was that structural damage (kidney biopsy?) or similar to presented functional = Fanconi syndrome?

Reviewer 2 Report

This a very interesting presentation of a child with de novo mt-DNA mutation with clinical manifestation of Fanconi syndrome showing retardation of growth during first 5-year period.

There are some points to be clarified and improved.

  1. It should be more clearly stated why the authors considered growth retardation as something unusual. They clearly state in the introduction that it is a trait of the Fanconi syndrome. In my opinion it is just cause by tubulopathy, that may be diagnosed with delay.
  2. The abstract is not informative for the reader.  
  3. Thera are some useless fragments in the case presentation. Like homocysteine, amylase level.
  4. The pedigree chart should be removed. The statement of the lack of DNA mutation in family members is sufficient.
  5. A large genetical part of discussion is redundant. It is enough to state that it was a de novo mutation in the family and stress a higher rate of mutated DNA in urine sample.
  6. The presented table with finding of other authors is nice. Perhaps you don’t need to use abbreviation in the table.

Round 2

Reviewer 2 Report

The authors have improved the paper, however some changes are necessary.

1.The authors should considered the modification of the title - Growth retardationan in the course of Faconi syndrome caused by the 4977- 2 bp mitochondrial DNA deletion: a case report

2. line 15 of the abstract 'Fanconi syndrome and growth retardation' please concide a change to 'growth retardation in the course of Fanconi syndrome'

3. line 23 of the abstract 'This was the first report to link Fanconi syndrome and growth retardation as the initial major clinical manifestation' please change to 'This is a report of growth retardation as the initial major clinical presentation of Fanconi syndrome'

4. Please modify in similar way the concusion section after discussion.
